# Variants of NAV3, a neuronal morphogenesis protein, cause intellectual disability, developmental delay, and microcephaly

Amama Ghaffar[1,2,20], Tehmeena Akhter ⬧[1,2,20], Petter Strømme[3], Doriana Misceo[4], Amjad Khan[5,6,7], Eirik Frengen ⬧[4], Muhammad Umair[8], Bertrand Isidor[9], Benjamin Cogné[9], Asma A. Khan[2], Ange-Line Bruel[10], Arthur Sorlin[10,11], Paul Kuentz ⬧[10], Christine Chiaverini[12], A. Micheil Innes[13], Michael Zech[14,15,16], Marek Baláž[17], Petra Havrankova[18], Robert Jech[18], Zubair M. Ahmed ⬧[1], Sheikh Riazuddin[19] & Saima Riazuddin ⬧[1] ✉

Microtubule associated proteins (MAPs) are widely expressed in the central nervous system, and have established roles in cell proliferation, myelination, neurite formation, axon specification, outgrowth, dendrite, and synapse formation. We report eleven individuals from seven families harboring predicted pathogenic biallelic, de novo, and heterozygous variants in the *NAV3* gene, which encodes the microtubule positive tip protein neuron navigator 3 (NAV3). All affected individuals have intellectual disability (ID), microcephaly, skeletal deformities, ocular anomalies, and behavioral issues. In mouse brain, *Nav3* is expressed throughout the nervous system, with more prominent signatures in postmitotic, excitatory, inhibiting, and sensory neurons. When overexpressed in HEK293T and COS7 cells, pathogenic variants impaired NAV3 ability to stabilize microtubules. Further, knocking-down *nav3* in zebrafish led to severe morphological defects, microcephaly, impaired neuronal growth, and behavioral impairment, which were rescued with co-injection of WT *NAV3* mRNA and not by transcripts encoding the pathogenic variants. Our findings establish the role of *NAV3* in neurodevelopmental disorders, and reveal its involvement in neuronal morphogenesis, and neuromuscular responses.

Microtubules plus-end tracking proteins (+TIPs) are a group of microtubule associated proteins (MAPs) well known to be involved in neurodevelopmental and neurodegenerative disorders[1]. These multifunction regulatory proteins interact with microtubules to direct cell division, protein trafficking, signal transduction, and cell polarity for migration and morphogenesis[2]. +TIPs harbor a subclass, neuron navigating proteins (NAVs) having SxIP motif (Ser/Thr:X: Ile/Leu: Pro) that interact with end-binding (EB) proteins and alpha-tubulin subunits on growing microtubules[3]. Structurally NAVs harbor calponin homology domain (CH) and microtubule-binding domain (MTBD) that facilitate binding with actin and microtubules, respectively, along with diverse cellular Activities (+AAA) domain responsible for ATPase type activity[4].

In vertebrates, three NAV proteins are known: NAV1, NAV2, and NAV3, which are homologs of *un-53 C. elegans* protein involved in cell migration, neurite growth, and axonal elongation as mutants showed deficits of neuronal outgrowth along with egg-laying defects[5]. *Nav1* has high expression in the murine heart and nervous system, specifically in postmitotic and promigratory neurons[6]. However, *Nav3* expression declines after birth, emphasizing the fact that it is an early development protein required for axonogenesis and synapse formations[6]. Similarly, NAV2 also has high expression in the central nervous system (CNS) and participates in brain development through its interaction with actin via ABI and ARP2/3 complex[7]. Loss of NAV2 caused cerebral hypoplasia and hypoactivity in mice[8]. Finally, *Nav3* longest isoform (~10 kb) is expressed in the nervous system as early as 10 days post-conception in murine[9]. In the neurons, NAV3 participates in cytoskeleton organization via binding with actin and MAPs, such as Ab1, p73, and Src[10]. In zebrafish, *nav3* shows high expression in the brain, somite, liver, heart, swim bladder, and intestines, and knocking out leads to cardiogenesis

deficits and a decreased survival rate[11]. While *nav3a* zebrafish morphants have defective cellular movements at the liver budding stage and caused an abnormal organogenesis[12]. In short, all the three NAVs show involvement in neuronal migration through expression-based studies.

Recently, through transmission disequilibrium test protein-truncating variants in *NAV3* have been identified in individuals with autism spectrum disorder (ASD) and other neurodevelopmental disorders (NDDs)[13]. However, no functional studies were conducted to determine their impact on the NAV3 function. Here, we report seven families showing vast genetic and phenotypic spectrum associated with *NAV3* variants. The enrolled individuals of multiple ethnicities show intellectual disability (ID), microcephaly, speech delay, aggressive behavior, and skeletal deformities as the most common phenotypes. with multiple patterns of inheritance, including autosomal recessive, autosomal dominant, and de novo.

## Results

### Genetics and clinical studies

After the identification of candidate variants in *NAV3* in the affected individuals of family PKMR471 from Pakistan, using GeneMatcher services, we subsequently connected with colleagues worldwide[14]. Together, we report seven unique *NAV3* variants in eleven affected individuals from seven unrelated families (Fig. 1a). All the identified individuals have ID (HP:0010864) as a common phenotype varying from mild to severe (Fig. 1b; Table 1). Almost all the affected individuals for whom clinical data is

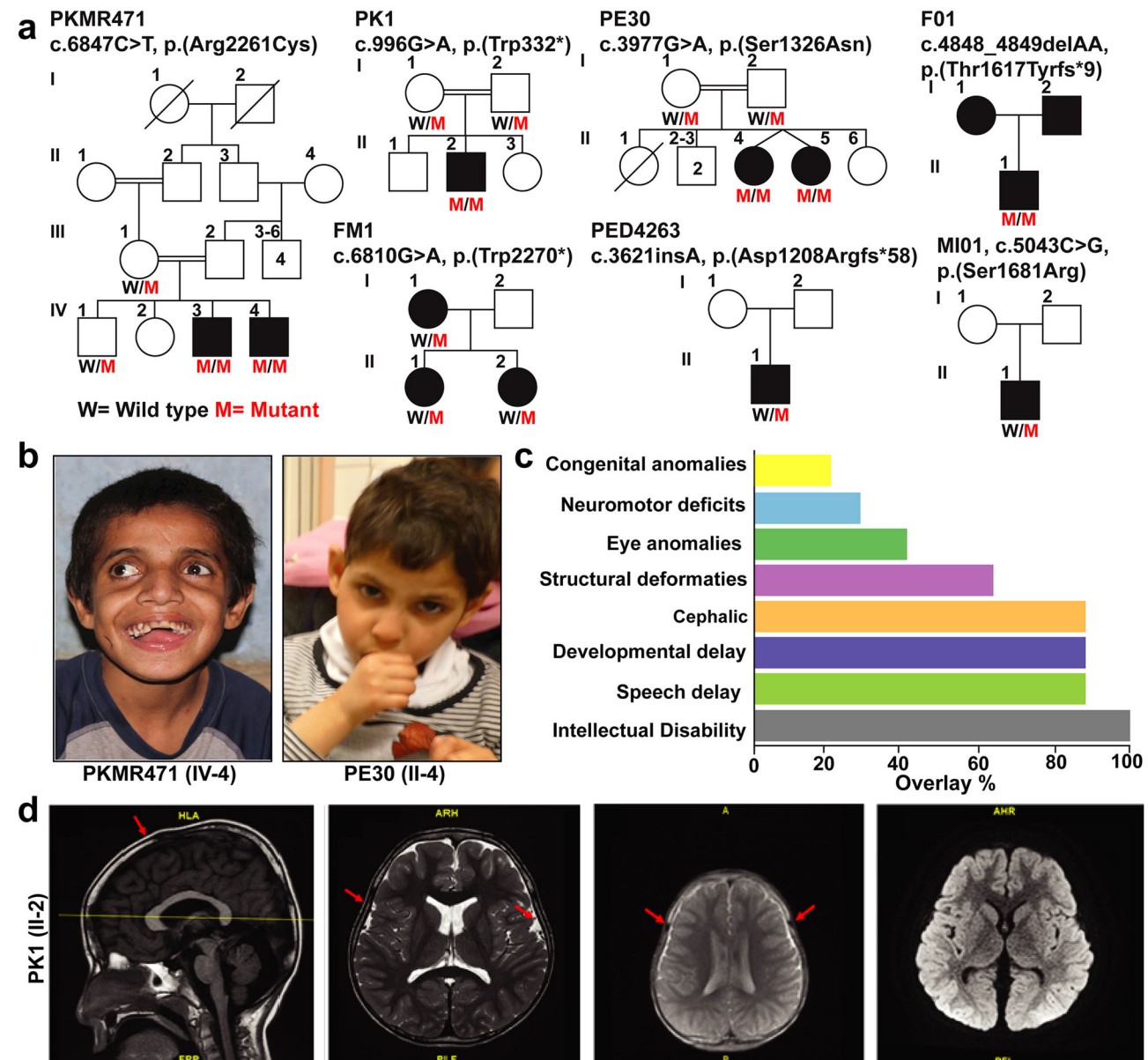

**Fig. 1 | *NAV3* genetic variants cause intellectual disability, microcephaly, speech, and development delay in humans. a** Pedigrees of seven families from diverse ethnicities with affected individuals due to genetic variants in *NAV3* with multiple inheritance patterns. Filled symbols represent affected individuals, while a double horizontal line connecting parents shows a consanguineous marriage. Affected individuals of PKMR471, PK1, PE30, and F01 families are homozygous for the *NAV3* variants. In contrast, family FM1 has a dominant inheritance pattern, and affected individuals of families PED4263 and MI01 have de novo variants. **b** Representative images of facial features of PKMR471 affected individual (IV:3) with severe ID showing dentofacial dysmorphism, microcephaly, hypertelorism, and strabismus, while affected individual of family PE30 (II:4) with moderate ID has microcephaly and skin pigmentation problems. **c** Bar graph representing frequency salient phenotypic features observed in the individuals that harbor genetic variants in *NAV3*. ID, speech, and developmental delay, and cephalic are commonly shared features. **d** MRI of the control and an affected individual of family PK1. Similar to control, the overall brain morphology of the affected individual was also unremarkable. However, the affected individual had mild delayed myelination (marked by red arrows) represents mild delayed myelination. Two minor developmental venous anomalies through the right side of the midbrain and cerebellum.

Table 1 | Phenotypic characteristics of individuals harboring NAV3 variants

| Family | PKMR471 | PE30 | MI01 | F01 | PED4263 | PK1 | FM1 | | |
|---|---|---|---|---|---|---|---|---|---|
| Variant | p.(Arg2261Cys) | p.(Ser1326Asn) | p.(Ser1681Arg) | p.(Thr1617Tyrfs*9) | p.(Asp1208Argfs*58) | p.(Trp332*) | p.(Trp2270*) | | |
| Genotype | Homozygous | Homozygous | De novo | Homozygous | De novo | Homozygous | Heterozygous | | |
| Ethnicity | Pakistani | Pakistani | - | Caucasian | French | Pakistani | European | | |
| Individual | IV:3 | II:4 | II:1 | II:1 | II:1 | II:2 | I:1 | II:1 | II:2 |
| Sex | Male | Female | Male | Male | Male | Male | Female | Female | Female |
| Current Age/age of evaluation (years) | 22 | 19 | NA | 8 years | 4 | 12 | 50 | 30 | 27 |
| Age of onset | By birth | By birth | NA | ND | By birth | By birth | ND | ND | 14 |
| Intellectual disability | Severe | Moderate | Mild | Mild | Mild | Severe | Mild | Mild | Mild |
| Speech delay | Yes, Speak few words 3 years | Yes, Few words at the age of 5 years | Yes, First word at 2 years | Yes, First word at the age of 3 years | Yes | Yes, Two to three words at 2 years of age | Preserved Speech | Preserved Speech | Preserved Speech |
| Developmental delay | Yes, Speech and walk delay | Yes, Walk at 2 years of age | Yes, Walk at 2 years of age | Yes | Yes | Yes, Still can't, walk only with aid Cannot self-feed | ND | ND | ND |
| Head abnormalities | Microcephaly | Microcephaly | Microcephaly | +1.4 SD | +2 SD | Microcephaly | No | No | No |
| Structural abnormalities | Yes, dentofacial abnormality, curved backbone, hypertelorism, and feet deformity | Yes, dentofacial abnormality, curved backbone, hypertelorism, and feet deformity | Yes | Short hands, Hypertelorism, long palpebral fissures | Pointed chin | No | No | No | No |
| Behavioral problem | Aggressive | Aggressive | Hyperactivity with stereotypic hand movements | ADHD | Frustration, ADHD | Aggressive Introvert | No | No | No |
| Eyes anomalies | Hypertelorism and mild strabismus | Hypertelorism and strabismus (Exotropia) | No | Strabismus (hypermetropia) large eyebrows | No | No | No | No | No |
| Ear anomalies | No | No | No | Large ear lobules | Protruding ears | No | No | No | No |
| Heart defects | No | No | No | NA | Ventricular septal defect at birth spontaneously closed | No | No | No | No |
| Neuro/muscular | Yes, cannot walk without support First walk attempt 8 years of age | Yes, cannot walk without support First walk attempt at 4 years of age | NA | Na | No | Yes, cannot walk without support | Generalized dystonia, wheelchair-bound | Mild dystonia (face) | Generalized dystonia, dystonia of neck, face, all 4 limbs, walking with support only |
| Congenital anomalies | No | No | Yes | testicular ectopia | No | | not reported | not reported | not reported |
| Other | Hypotonia | Pigmentary skin changes, normal CGH- array, MECP2 and UBE3A expression normal, EEG normal | NA | NA | Large, crumbled café-au-lait stain on chest and back | Hypotonia | Movement disorder (dystonia) | Movement disorder (dystonia) | Movement disorder (dystonia) |

NA no information available.

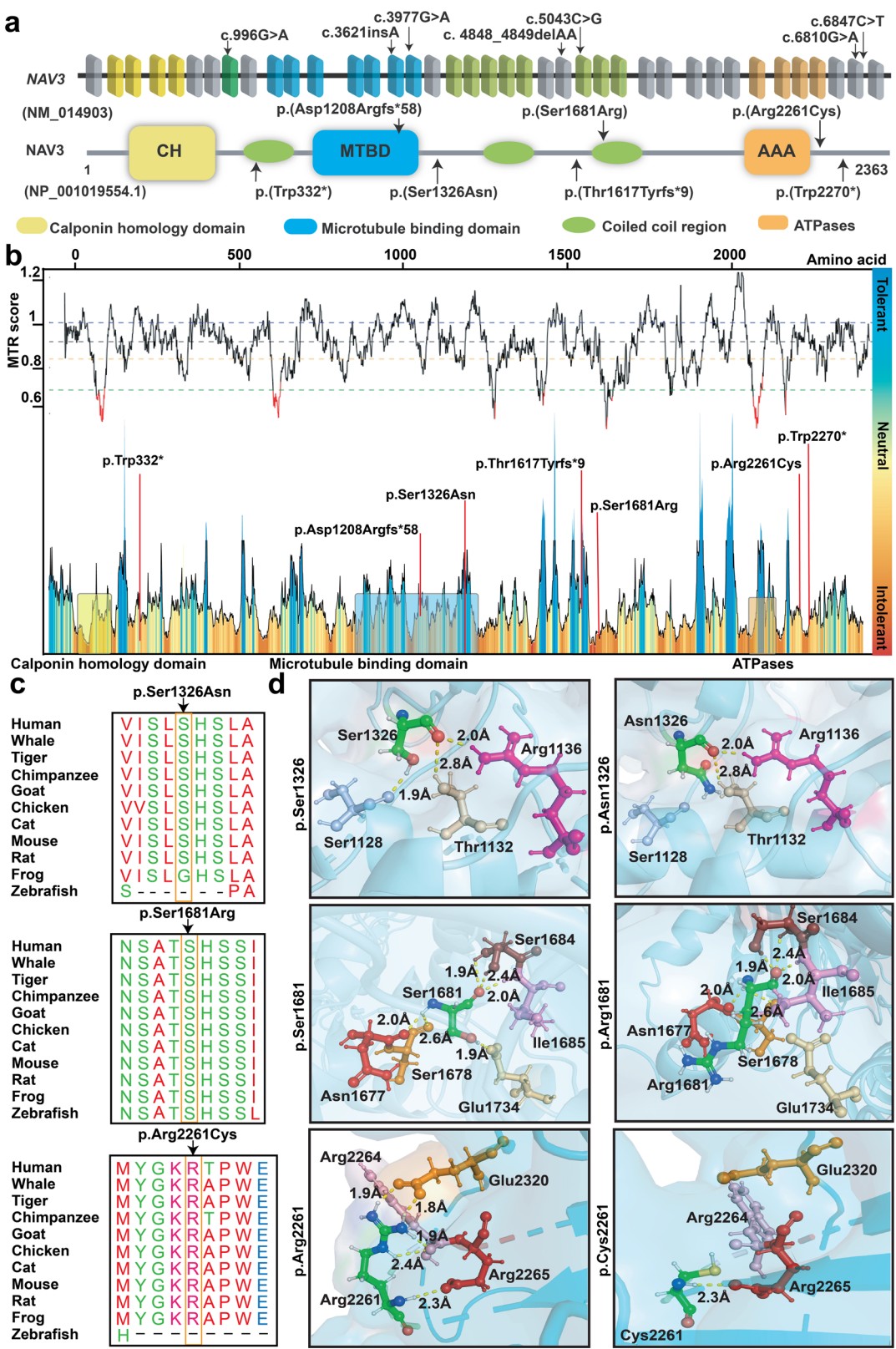

available show speech delay (HP:0000750), global developmental delay (HP:0001263), and microcephaly (HP:0000252) (Fig. 1c). Other clinical features include aggressive behavior (HP:0000718) neuromuscular issues (HP:0001252, HP:0001270, HP:0000486, and HP:0001332), dysmorphic structural abnormalities (HP:0000164), hypertelorism (HP:0000316), and congenital abnormalities (HP:0000035) (Fig. 1c; Table 1). Magnetic resonance

imaging of the individual II:2 from family PK1 reveals normal brain morphology, including brainstem, corpus callosum, sella, and suprasellar structures along with preserved gray-white matter. However, two small developmental venous anomalies through the right side of the midbrain and cerebellum were observed with slightly delayed myelination as compared to the patient's age (Fig. 1d).

**Fig. 2 | *In-silico* analysis and 3D protein modeling support deleterious impact of NAV3 variants. a** Schematic representation of human *NAV3* gene (top) and encoded protein (bottom) structures, along with the variants identified in this study. *NAV3* in humans is comprised of 39 coding exons, while encodes Neuron Navigator 3 (NAV3) protein with multiple functional domains, including the Calponin homology region (CH), microtubule-binding domain (MTBD), ATPases (AAA), and three coiled-coil regions (green ovals). **b** The Missense Tolerance Ratio (MTR) graph and Intolerance landscape visualization of NAV3 via MetaDome are presented with relative positions of the variants identified in our cohort. All of the identified variants are located in relatively intolerant regions of the NAV3. **c** High conservation of the residues, replaced due to the missense variants found in this study, was observed during evolution. **d** Three-dimensional (3D) protein modeling illustrating an overview of NAV3 protein. The left panels represent wild-type residues at given positions, while their alternate mutated residues are in the right panels. Color code: green element; targeted amino acid; interacting amino acids: mustard, pink, purple, red, badge, rose, and burgundy, while hydrogen bonding between the residues is shown with yellow dotted lines along with the distances in Å. For p.Ser1326Asn substitution, a loss of hydrogen bond with p.Ser1128 (purple) was predicted. Similarly, for the p.Ser1681Arg variant, loss of hydrogen bond with p.Glu1734 (badge) was predicted. The p.Arg2261Cys variant is predicted to abolish four hydrogen bonds: two with p.Arg2264 (pink) and two with p.Glu2320 (mustard).

All probands of the identified families were exome sequenced at participating centers, followed by Sanger sequencing to confirm the variants and their segregation. Among the identified variants in *NAV3*, four have recessive inheritance patterns (p.(Arg2261Cys), p.(Trp332*), p.(Ser1326-Asn), p.(Thr1617Tyrfs*9)) in families PKMR471, PK01, PE30, and F01 respectively (Fig. 1a). Affected individuals of families MI01 and PED4263 have de novo variants (p.(Ser1681Arg); p.(Asp1208Argfs*58)), while the dominant inheritance of p.(Trp2270*) is observed only in one family FM1 (Fig. 1a). These variants are spread all over the protein (Fig. 2a). In-silico, pathogenicity predictors show that variants identified in our cohort impact the protein function and are pathogenic (Table 2). The allele frequency for all variants is very low (<0.001%) in the general population (Table 2).

The probability of loss of function Intolerance scores (pLI) for the *NAV3* gene is 1, and Z constraint for missense variants is 1.33, showing the *NAV3* to be intolerant for any loss of function and missense variants[15]. The Missense tolerance ratio (MTR) scores and MetaDome for the missense variants show the identified variants to be slightly intolerant (Fig. 2b). Clustal Omega shows high conservation of all the ID-associated missense residue across various species (Fig. 2c). Next, we performed 3-Dimensional molecular modeling to further dissect the impact of identified missense variants on the protein folding and secondary structure (Fig. 2d). For hydrogen bond analysis PyMOL was used to find interacting bonds of WT and mutated amino acids to neighboring amino acids. The p.Ser1326Asn variant is predicted to cause a loss of hydrogen bond with p.Ser1128 residue due to small size and less hydrophobic nature of asparagine residue as compared to WT serine residue, and thus might impact protein secondary structure. The p.Ser1681Arg variant is also predicted to remove hydrogen bonding with p.Glu1734 residue due to less hydrophobic, smaller size and positive charge of arginine residue (Fig. 2d). Moreover, p.Ser1681Arg substitution causes energy destabilization (−0.0 kcal/mol). In contrast, the p.Arg2261Cys variant resulted in loss of four hydrogen bonds, two with p.Arg2264 and two with p.Glu2320 residues (Fig. 2d), thus likely impact protein folding and secondary structure.

## *NAV3* variants impact microtubule polymerization and structural stability

To functionally validate these bioinformatic findings, we transiently over-expressed WT and ID-variants harboring *NAV3* cDNAs in COS7 cells. NAV3 WT protein showed expression overlap with acetylated tubulin, a marker for polymerized microtubules with the formation of extended filopodia (Fig. 3). Similar to WT protein, the p.Arg2261Cys, p.Ser1326Asn, and p.Ser1681Arg variants harboring NAV3 also showed overlapping expression with stable microtubule structures along with condensed granule expression towards +Tips of microtubules (Fig. 3). However, the protein-truncating variants, except p. Asp1208Argfs*58, did not show +Tip granule formations (Fig. 3). Among the variants analyzed, NAV3 with p.Thr1617Tyrfs*9 and p.Trp332* had reduced or no bundling (Fig. 3) and show centralized expression in nucleus (not shown).

To further decipher the pathogenic impact of identified variants on the encoded protein, we studied microtubule stability in the presence of poly-merization inhibitor nocodazole. By exposing non-transfected HEK293T as well as COS7 cells to nocodazole, we observed an almost complete loss of stable microtubules and reduced expression of acetylated and tyrosinated tubulin (Fig. 4a, b and Supplementary Fig. 1a, b). Consistent with prior studies, GFP-NAV3^WT transfected cells showed detectable stable micro-tubules despite nocodazole treatment (Fig. 4a, b). Intriguingly, all of the identified ID-causing variants were not able to maintain the microtubule structures and had statistically significant reduced stability impact on polymerized microtubules when compared to NAV3^WT expressing cells (Fig. 4b, Supplementary Data 1), hence supporting their deleterious impact on the encoded NAV3.

## Knockdown of *nav3* in zebrafish affects neurodevelopment and behavior

RNAseq data for *Nav3* from the mouse organogenesis cells dataset shows high expression in neural tubes, postmitotic neurons, excitatory neurons, inhibitory neurons, and interneurons (Fig. 4c). Next, to determine the impact of the loss of NAV3 on brain development and structure, we turned to zebrafish and generated translational blocking (ATG_MO) morphants (Fig. 5a, b) on a *neuroD* transgenic zebrafish line (neuroD-GFP) genetic background, which expresses a GFP fluorescent marker through the transcriptional activity of *neuroD* promoters to progress neuronal differentiation[16]. Previous studies have shown that 10 ng of ATG_MO causes deficits in liver organogenesis[12], therefore, we synthesized the same ATG_MO as well as splice site-directed MO (SS_MO) for our studies (Fig. 5a). When injected at 1–2 cell stages, the *nav3* trans-lation blocking ATG_MO at 8 ng as well as 10 ng dose resulted in morphological, developmental deficits as compared to the control MO (scrambled nucleotides) group (Fig. 5b). Similar phenotypes were observed with the injection of 12 ng SS_MO (Fig. 5b). SS_MO injected morphants amplified cDNA as compared to control resulted into exon 2 skipping (c.del178_195), and deletion of 118 bp from *nav3* mRNA, leading to reading frameshift and premature truncation (p.Ile60Leufs*4) of the encoded protein (Fig. 5c). Based on the apparent abnormalities, we grouped the *nav3* morphants into four classes: (1) Severe: under-developed head and eyes, small body length, no swim bladder, yolk and heart edema, and curved tail embryos; (2) Moderate: small head size (microcephaly), yolk and heart edema with curved tail; (3) Mild: small head size, yolk, and heart edema; and (4) Normal: having no apparent difference from WT or control injected (Fig. 5b, d, e). Head-to-length ratios in *nav3* ATG morphants as compared to the control MO group further confirmed the microcephaly (Fig. 5f).

To gain further insights into neuronal growth patterns, we performed confocal imaging of the brain of control and *nav3* ATG_MO injected morphants. In contrast to control injected fish, the *nav3* ATG_MO injected morphants did not show any apparent expressional change in telencephalon regions except habenula (Hb) reduced size. However, the midbrain and hindbrain regions showed the absence of Torus longitudinalis (TL) with no mid region expression of neurod as well as reduced neuronal patterns and size of optic tectum and cerebellum (Fig. 5g, h). Further, to assess the impact of observed developmental abnormalities on motor responses in mild and normal class morphants, we subjected the singly housed larvae to 5 min of dark followed by 5 min in light[17]. We first observed no movement differences between un-injected and control_MO (10 ng) zebrafish embryos, while significant decreased (p < 0.0001) movements were observed in *nav3* ATG_MO as compared to controls. (Fig. 5i).

**Table 2 | In-silico pathogenicity prediction analysis of NAV3 variants**

| Pathogenic predictors | p.(Arg2261Cys) | p.(Ser1326Asn) | p.(Ser1681Arg) | p.(Thr1617Tyrfs*9) | p.(Asp1208Argfs*58) | p.(Trp332*) | p.(Trp2270*) |
|---|---|---|---|---|---|---|---|
| gnomAD | 0.0000201 | NDA | NDA | NDA | NDA | NDA | NDA |
| CADD | 34 | 25.2 | 26.1 | NA | NA | 41 | 54 |
| MTR | 0.9 | 0.87 | 0.86 | NA | NA | NA | NA |
| MetaDome | Slightly tolerant | Slightly intolerant | Slightly intolerant | NA | NA | NA | NA |
| GERP++ | 5.4 | 5.96 | 4.49 | NA | NA | 5.5 | 5.4 |
| M-CAP | Damaging | Tolerated | Damaging | NA | NA | NA | NA |
| SIFT | Tolerated | Damaging | Damaging | NA | NA | NA | NA |
| MutationTaster | Disease causing | Disease causing | Disease causing | NA | NA | Disease-causing | Disease-causing |
| Provean | Pathogenic | Uncertain | Pathogenic | NA | NA | NA | NA |
| Polyphen-2 HumDiv | Probably Damaging | Probably damaging | Probably damaging | NA | NA | NA | NA |
| ACMG classification | BP1[a], BP4[b], PM2[c] | BP1[a], BP4[b], PM2[c] | BP1[a], BP4[b], PM2[c] | PVS1[d], PM2[c] | PM2[c] | PVS1[d], PM2[c] | PM2[c] |

NDA no data available, NA not applicable.
[a]BP1 = Missense variant in a gene for which primarily truncating variants are known to cause disease. (Benign, Supporting).
[b]BP4 = MetaRNN = 0.415 is between 0.267 and 0.43.
[c] PM2 Variant not found in gnomAD genomes.
[d]PVS1 = Null variant (nonsense) in gene NAV3, predicted to cause NMD.

Importantly, we were able to significantly improve (****$p < 0.0001$) the phenotypes of nav3 morphants with human $NAV3^{WT}$ mRNA in a dose dependent manner (Fig. 6a), injected at 1–2 cell stages. As compared to WT, mRNA injected groups with NAV3 variants showed significantly reduced (****$p < 0.0001$) abilities to rescue morphant phenotypes (Fig. 6b), thus further confirming their pathogenic nature. In summary, the in vivo study using zebrafish as a knockdown model for nav3 demonstrated the role of NAV3 in neuronal morphogenesis in mid and hindbrain regions.

## Discussion

In this study, we describe a cohort of patients harboring deleterious variants in the NAV3 gene. The affected individuals show ID, microcephaly, skeletal deformities, ocular anomalies, and behavioral issues besides other clinical symptoms. Recently, another study documented nine nonsense, 12 frameshifts, and three splicing sites predicted pathogenic variants of NAV3 in subjects with ASD and NDDs (Fig. 7) based on transmission disequilibrium test[13]. ASD in about half of their cases coexisted with ID (18/35) and attention-deficit/hyperactivity disorder (ADHD; 15/35)[13]. Among our cases, ADHD was observed in subjects with truncating variants (Table 1). In contrast to almost all truncating variants found in ASD cases, half of our subjects with ID had missense variants (Fig. 1). Apparently, missense variants (presumably hypomorphic alleles) of NAV3 cause ID, while truncating alleles are responsible for ASD with additional neurological findings. In general, phenotype spectrum observed in our cohort overlaps with the prior study, and thus further confirm the role of NAV3 in the brain development and cognitive function.

In contrast to prior study, which mostly report de novo variants, among our cases, we observed three different allele patterns, autosomal recessive, dominant, and de novo. Although our sample is not large enough for a meaningful genotype-phenotype correlation, intriguingly, individuals with biallelic NAV3 variants had severe ID with the prevalence of other features such as microcephaly, skeletal deformities, and behavioral problems. Affected individuals with heterozygous alleles (dominant and de novo cases) had mild ID with less prevalence of subsided issues (Table 1), which might reflect haploinsufficiency. However, larger variant sets and functional studies would be required to classify disease mechanisms of variants and to devise clinically applicable genotype-phenotype correlations.

NAV3 expression analysis in scRNA Organogenesis Cell Atlas revealed dispersed signatures in the neuron progenitor cells. However, higher expression was observed in the postmitotic neurons as early as E16.5 majorly in forebrain and midbrain, which supports potential role in cell migration and axonal guidance for the neurons to reach their functioning regions and connectivity points. These processes when interrupted cause morphogenesis related disorders such as agenesis of corpus callosum, microgyria, and agyria which ultimately lead to ID, microcephaly, developmental delays, poor muscle tone, and motor function[18–21]. We observed similar deficits in our cohort (Table 1). Taken together, our findings support the pivotal function of NAV3 in neuronal morphogenesis and neurological disorders in humans.

NAV3 also interacts with cytoskeletal microtubules, which are heterodimer structures made up of alpha and beta tubulin. In initial development stages, neurons begin as unpolarized spherical-shaped cells with microtubules spreading out from organizing centers and form the basis of the γ-tubulin complex, which further act as templates for α- and β-tubulin dimers to begin polymerization for neurite formation, axon specification, elongation and branching along with dendrite and synapse formation[11,22,23]. NAV3 overexpression studies showed unpolarized cells to form filopodia extensions with granular expression at +Tip of microtubules. In contrast to prior study that reported expression only at comets of +Tips[24], we found overexpressed NAV3 overlapped with microtubule bundles (Figs. 3, 4). NAV3 is also known to provide polymerization stability to microtubule dynamics[4,24], a function we also observed to be significantly reduced in the ID-causing variants in our heterologous cells assay (Fig. 4a, b), supporting their pathogenic nature.

In our zebrafish nav3 knockdown model, we observed major developmental deficits, including heart malformation and microcephaly (Fig. 5b), similar to zebrafish nav3 knockouts[11]. We observed generalized microcephaly in nav3 morphants as well. The major brain structures

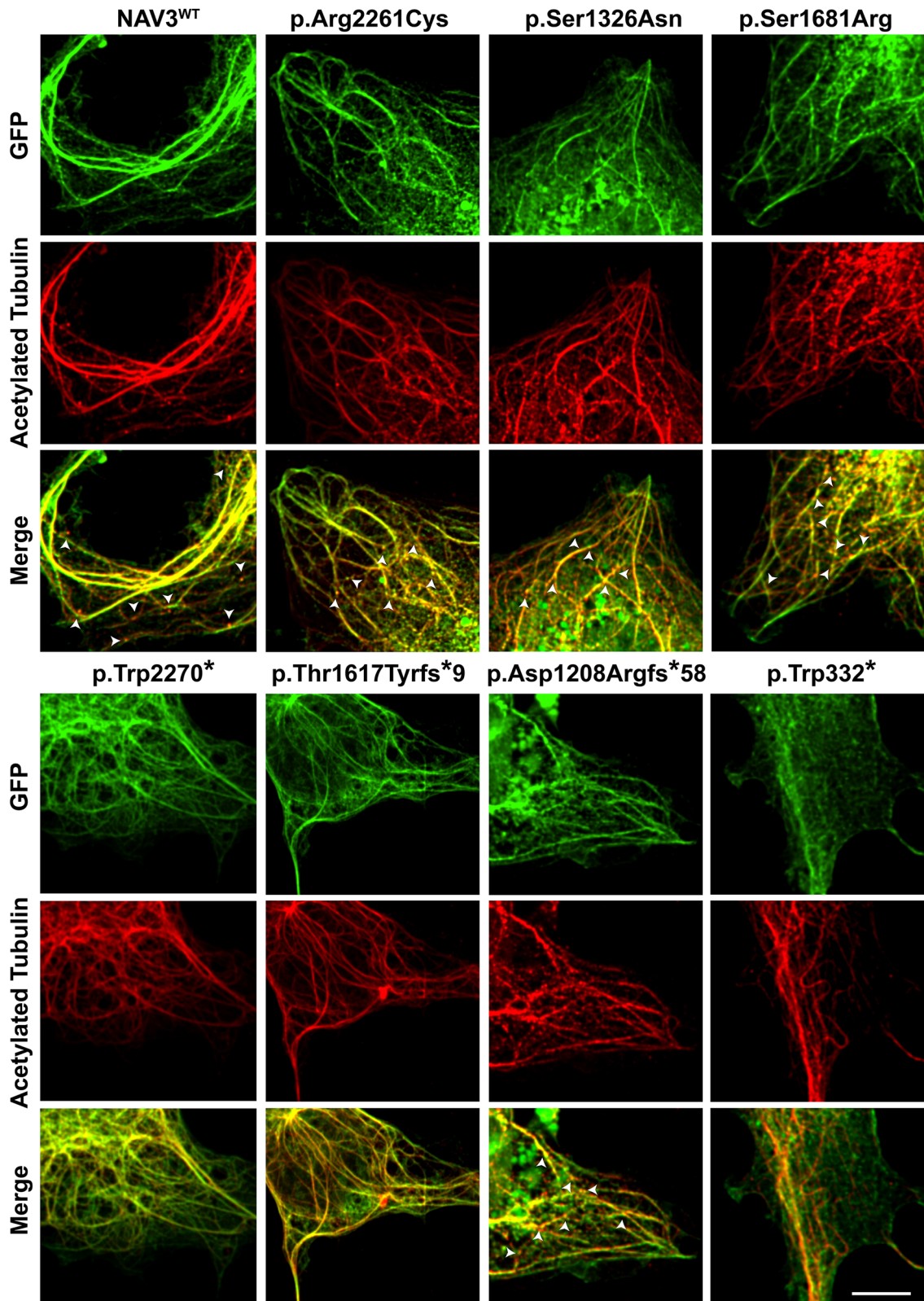

**Fig. 3 | NAV3 interacts with microtubules in COS7 cells.** GFP-tagged WT and mutated NAV3 proteins (green) showed an overlap expression (yellow) with microtubules, decorated with acetylated tubulin antibodies (red). Granular expression of NAV3 was observed at dendrite-like structure formations and tip ends of microtubules. White arrow heads show the granule type expression at +Tip ends of microtubules. Similar to WT protein, the p.Arg2261Cys, p.Ser1326Asn, and p.Ser1681Arg variants harboring NAV3 also showed overlapping expression with stable microtubule structures. However, the protein-truncating variants, except p. Asp1208Argfs*58, did not show +Tip granule formations. Among the variants analyzed, NAV3 with p.Thr1617Tyrfs*9 and p.Trp332* had reduced or no bundling and show centralized expression in nucleus. Scale bar: 20 um.

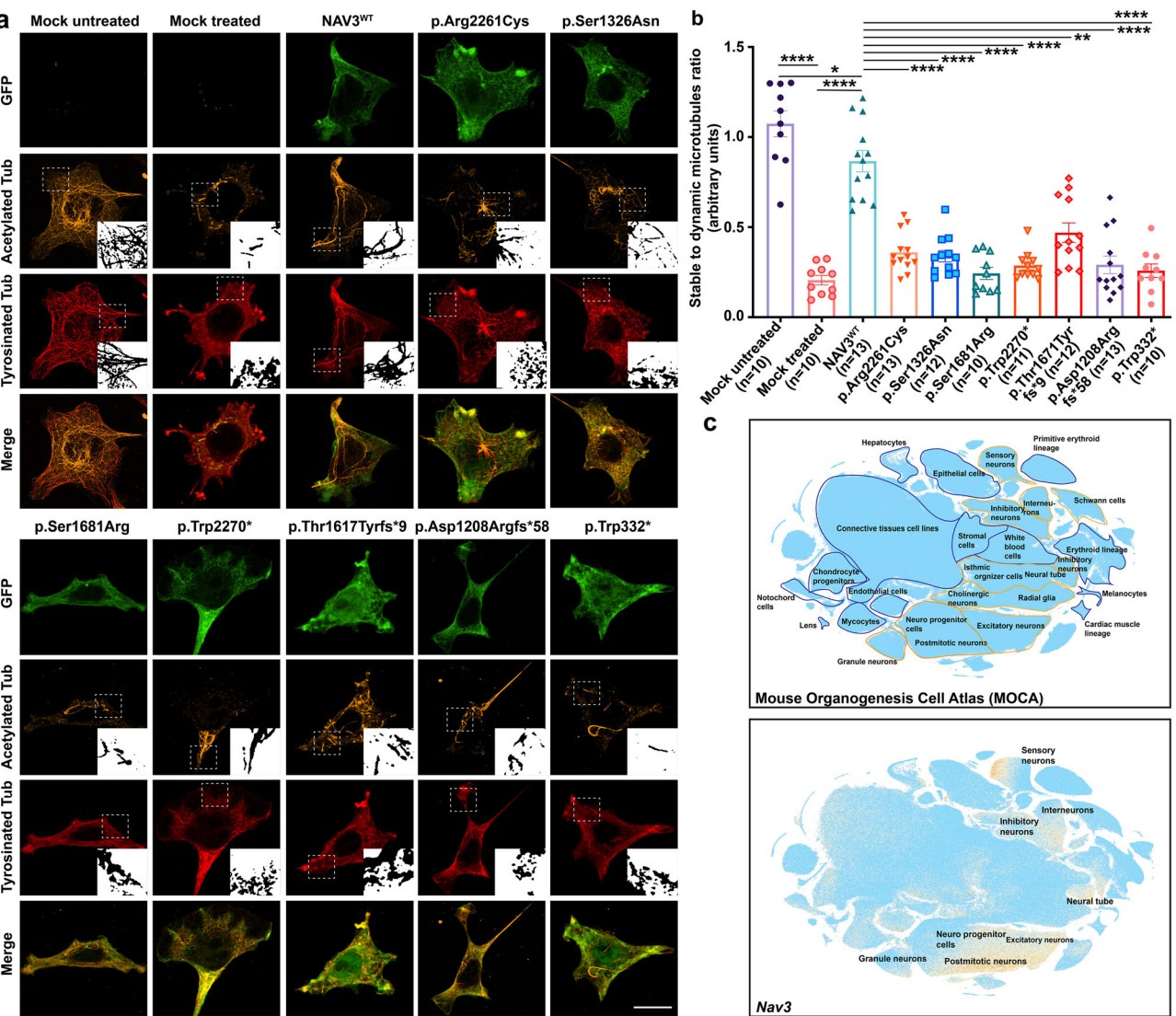

**Fig. 4 | ID-associated variants impact NAV3 ability to stabilize microtubule in HEK293T cells. a** Representative confocal images of GFP-NAV3 wild type (WT) and mutated proteins (green) overexpressing HEK293T cells treated with 10 μM nocodazole for 1 h, and immuno-labeled with microtubule acetylated tubulin (orange) and tyrosinated tubulin (red) markers. Compared to cells over-expressing WT GFP-NAV3 with a stable microtubular network despite the nocodazole treatment, all the ID-associated variants impact the microtubule-stabilization function of NAV3. Dotted white lines show zoomed in areas in insets. Scale bar: 20 μm. **b** Bar graph quantification ratio of stable to dynamic microtubules for all respected treated groups. At least 10 transfected cells per construct were imaged and quantified. All the variants expressing cells results were compared to WT using the paired student *t-test*. NAV3 proteins harboring ID-associated variants showed significantly reduced microtubules stability (**** $p < 0.0001$; *$p < 0.0350$; **$p < 0.00220$). Error bars represent standard error of the mean. **c** Single-cell RNA (sc-RNA) expression profile of mouse cell clusters at early organogenesis (top panel) generated from transcriptomes of around 2 million cells derived from 61 embryos staged between 9.5 and 13.5 days of gestation (data available from UCSC cell browser). The transcriptome data (bottom panel) shows the highest expression of *Nav3* in the neural tube and postmitotic neurons, while expression also observed in inhibitory and inter neurons. Dispersed expression in neuronal progenitor cells were also observed.

morphologically impacted in *nav3* morphants were (a) habenula (Hb), which contains habenular neuronal cell types (Hb01-Hb15); (b) torus longitudinal (TL); and (c) Optic tectum (OT) carry mostly granule cells with multiple neuronal subtypes, including mostly GABAergic neurons; and (d) cerebellum, which harbor Purkinje cells, eurydendroid cells, and granule cells with long axon terminals and dendrites. All these regions of brain are functionally interconnected for coordinated motor responses. Thus, reduced sizes of these major structures among *nav3* morphants might impact the neuronal communication among different parts of the brain involved in motor responses. Further research in murine models with conditional deletion of *Nav3* in specific brain regions would aid in determining the precise role of NAV3 in neuronal subtype proliferation, patterning, connection, and communication.

The light-based stimulation analysis shows low movements of *nav3* morphants (Fig. 5i), which could be attributed to one or more of the following: (a) generalized developmental and structural abnormalities; (b) brain developmental deficits, or (c) heart deficits and yolk edema. The optic tectum is center of neurons receiving signals from the retina and torus longitudinalis for vision-based responses[25], and low movement response in *nav3* morphant might be due to structural or connectivity deficits in these regions of the brain. However, our light-based simulation analysis is only empirical data, and cannot be used to define cognition function. Future studies in *Nav3* murine model including behavioral assays, neuronal connectivity, and physiological measurement would help in defining the precise role of NAV3 in cognitive function.

In conclusion, taken together with prior studies, biallelic and monoallelic variants of *NAV3* are responsible for a spectrum of NDDs with clinical features including ID, microcephaly, global developmental delay, or autism. Our findings further substantiate association of *NAV3* variants with NDDs in humans.

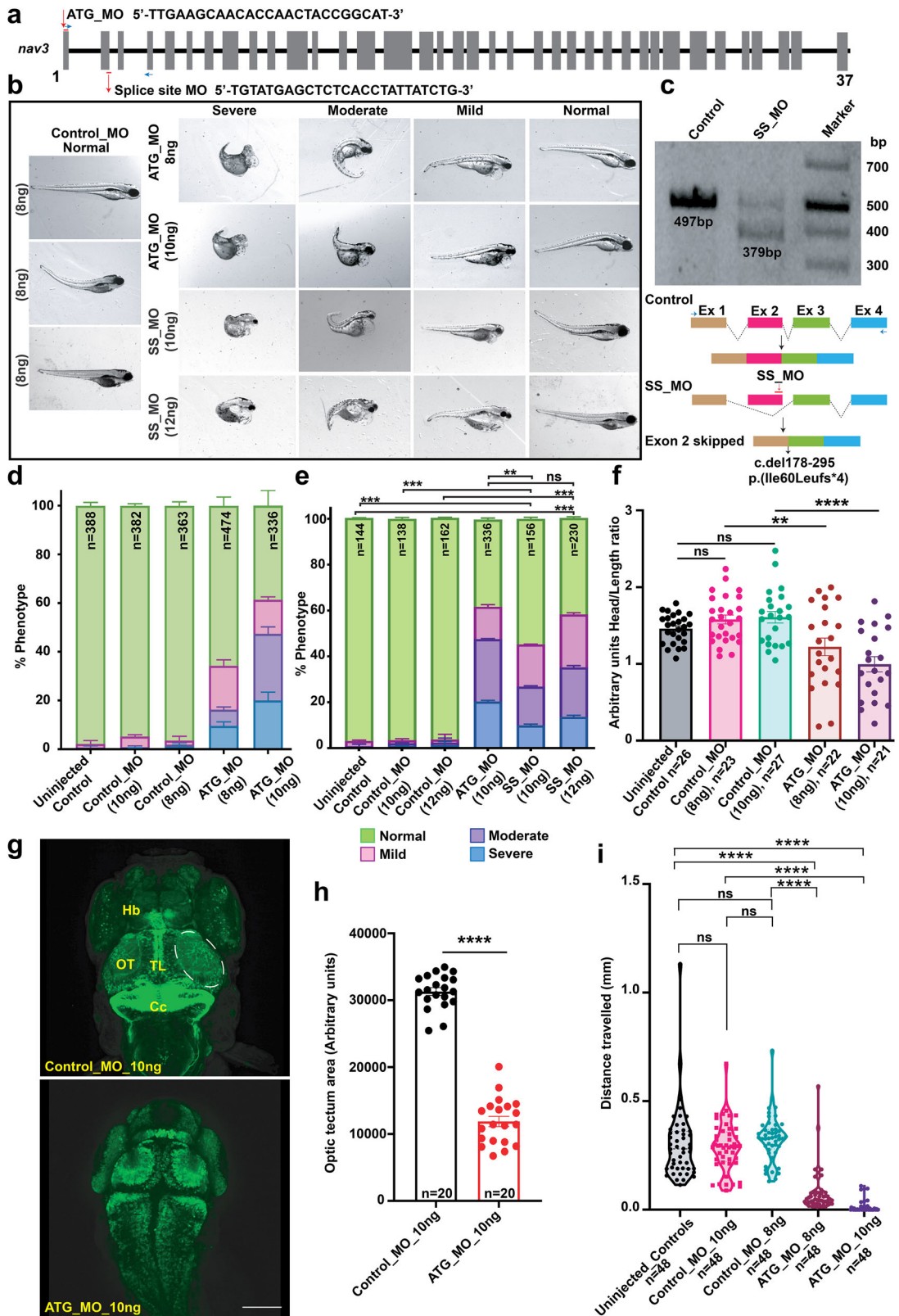

### Subject enrollment and clinical evaluation

Written informed consent was obtained from all individuals involved. This study adheres to the World Health Association Declaration of Helsinki and was approved by the Institutional Review/Ethic Boards of all the participating institutes. Medical and family history, developmental childhood milestones, anthropometric measurements, and findings of physical examination were collected, and detailed clinical phenotypes were described using Human Phenotype Ontology terms[26]. Venous blood samples were collected from research participants for DNA extraction.

**Fig. 5 | Zebrafish *nav3* morphants show severe development abnormalities.**
**a** Schematic representation of zebrafish *nav3* gene along with position and sequence
of ATG translational blocker (ATG_MO) and splice site (SS_MO) morpholinos.
**b** Representative images of control (Control_MO), *nav3* ATG_MO (10 ng), and
SS_MO (12 ng) morphants. Based on their developmental morphology, *nav3*
morphants were grouped into severe, moderate, mild, and normal classes. Severe
class morphants had underdeveloped heads and eyes, small body length, missing
swim bladders, yolk and heart edema, and curved tails, while moderate class mor-
phants had small head size, yolk, and heart edema with curved tail in morphants. The
mild class morphants had small head sizes, yolk, and heart edema, while normal class
morphants had no apparent morphological difference from WT or control injected
larvae. **c** RT-PCR followed by gel electrophoresis revealed smaller PCR product in
SS_MO injected morphants (top). Sanger sequencing of the PCR products con-
firmed aberrant splicing (exon 2 skipping, deletion of 118 bp; schematically repre-
sented in bottom panel) due to the SS_MO injection in zebrafish larvae.
**d, e** Phenotype assessment-based bar graphs for control, *nav3* ATG_MO, SS_MOs.

**d** With increase in ATG_MO concentration, the phenotypic abnormalities among
injected larvae also escalated. **e** Compared to control, significantly reduced
(**$p < 0.0048$ and ****$p < 0.000123$) number of morphologically normal larvae
observed in ATG_ and SS_MOs injected morphants. **f** Head-to-length ratio bar
graph in uninjected, control_MO, and *nav3* ATG_MO morphants. Compared to
controls, *nav3* morphants have statistically significant reduced head sizes
(**$p = 0.006931$ and ****$p = 0.000027$, respectively), suggesting microcephaly.
**g** Confocal images of zebrafish brain from the *neuro-d* transgenic line, injected with
control (top) or *nav3* (bottom) morpholinos. Control_MO injected larvae show
normal morphology of optic tectum (OT), torus longitudinalis (TL), habenula (Hb),
and corpus cerebelli (Cc) regions. In contrast, *nav3* morphants had significantly
reduced (****$p < 0.000001$) OT (highlighted with white dashed lines), quantified in
(**h**). Scale bar: 100 um. **i** Violin plot showing statistically significant
(****$p < 0.000001$) decreased movement in *nav3* morphants as compared to con-
trols, when subjected to light stimulus. Error bars in (**d, e, f, h, I**), represent standard
error of the mean.

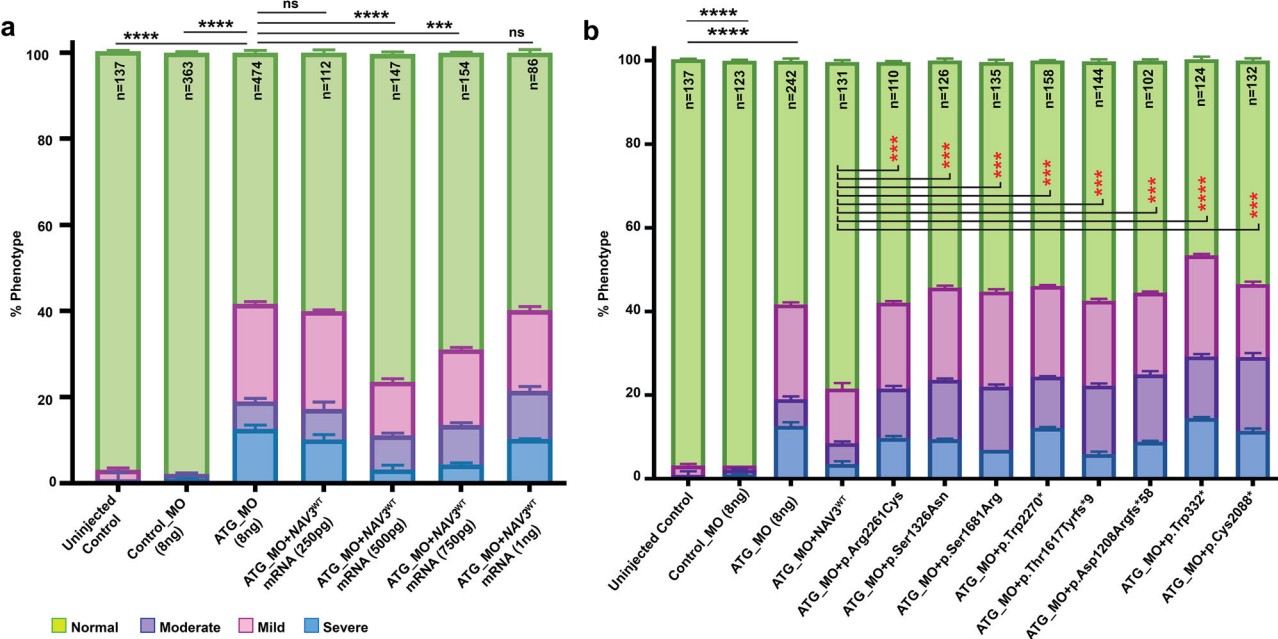

**Fig. 6 | *NAV3*-WT mRNA injections rescued the phenotype in morphants as**
**compared to variants harboring mRNAs. a** Bar graph representation of morphant
morphological phenotypes after injection of four different concentrations (250 pg,
500 pg, 750 pg, and 1000 pg) along with *nav3* ATG_MO. The 500 pg concentration
of NAV3-WT mRNA injection revealed the most significance rescue of normal class
larvae. NS not-significant; ***$p < 0.000250$; and ****$p < 0.00001$. **b** Rescue
experiments performed via co-injecting human *NAV3*^WT or variants harboring

mRNAs (500 pg) along with ATG_MO (8 ng) revealed significant rescue of normal
developmental class in WT mRNA injected morphants compared to ATG_MO
alone. The normal phenotype class in morphants from *NAV3*^WT and the variants
mRNAs co-injected were compared for statistical analysis (***$p < 0.0001$;
****$p < 0.00001$). Importantly, the ID-associated variant harboring *NAV3* mRNAs
could not rescue the normal developmental class of zebrafish, supporting their
pathogenic nature. Error bars in (**a, b**) represent standard error of the mean.

## Genetic analysis

Exome sequencing at various participating institutes was performed on
genomic DNA extracted from blood samples of enrolled subjects.
Data assembly, variant calling, and prioritization were performed as
reported previously[15]. Additional criteria of gene function and
expression associated with the central nervous system[25,27–32], was
added in ranking the final list of candidate variants for segregation
analysis.

## In-silico analysis and 3D protein modeling

Wild type NAV3 amino acid sequence from Uniprot (Q8IVL0-2) was used
as a reference for all in silico analyses. MTR scores and general intolerance
plot of NAV3 were generated using MetaDome software[33,34]. For analyzing
the evolutionary conservation of residues mutated in our cohort, Clustal
Omega was used to align NAV3 sequences from the UniProt database[35].
RNA expression profile for *Nav3* was analyzed using the University of

California Santa Cruz (UCSC) cell browser, having data for two million cells
derived from 61 embryos staged between 9.5 and 13.5 days of gestation
Mouse Organogenesis Cell Atlas[36,37].

For 3D protein-based structural analysis, the NAV3 sequence was
submitted to the I-Tasser website[38]. The predicted protein model with the
highest C score was further used for protein modeling to observe hydrogen
bond changes and structural impacts due to conserved amino acid sub-
stitutions using PyMol along with HOPE to assess variant differences in
accordance with wild-type amino acids[39,40].

## Expression constructs

The expression construct bio-EGFP-NAV3 for *NAV3* human cDNA was
generously gifted by Drs. Niels Galjart and ref. 24. Identified *NAV3* variants
were introduced via site-directed mutagenesis using QuikChange XL Site-
Directed Mutagenesis Kit from Agilent. All the constructs made were
validated using Sanger sequencing.

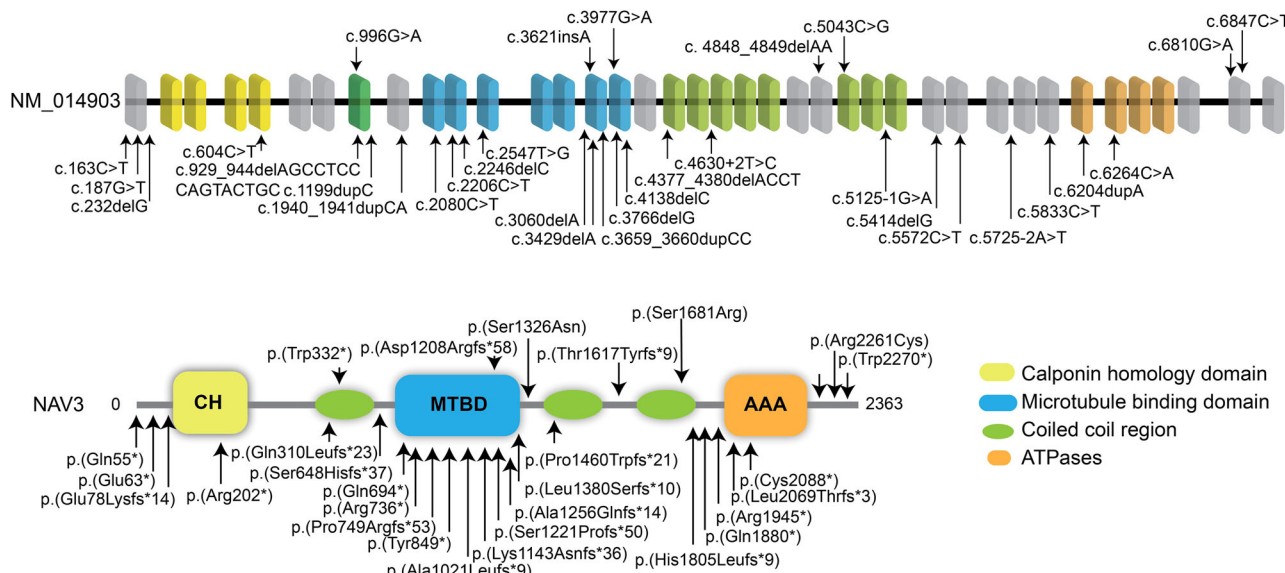

**Fig. 7 | *NAV3* variants cause a spectrum of Neurodevelopmental disorders (NDD) in humans.** Bar diagram of *NAV3* gene and protein structure showing mutations identified in this study at the top and mutations reported previously for Autism Spectrum Disorder (ASD) with other cooccurring NDDs at the bottom. Overall, among the 30 known variants, 10 are located in the MTBD domain, making it a suspectable hotspot region.

## Immunolocalization

COS7 (African green monkey kidney fibroblast-like) cell line was used to transiently express pbio-EGFP-*NAV3* WT and variants using Lipofectamine 2000 (Invitrogen Catalog no. 11668030). Cells were fixed after 24 h of transfection using 4% paraformaldehyde (PFA), followed by blocking for 1 h in 10% normal goat serum (NGS). To stain microtubules, primary acetylated tubulin mouse monoclonal antibody (1:500; Sigma Catalog no. T7451) was used along with Alexa 594 goat anti-mouse secondary antibody (1:500; Invitrogen Catalog no. A11032). DAPI (1:500) was used to label nuclei. After mounting, imaging was performed using a 60X oil immersion-based lens of a Nikon W-1 spinning disk at the iLabs core facility of the University of Maryland, Baltimore. Fiji software was utilized to process the images.

## Nocodazole assay

COS7 and HEK293T cells were transfected with various GFP-NAV3 constructs for 24 h, followed by 1-h treatment with 10 μM Nocodazole (Sigma Catalog no. M1404) in Dulbeco's Modified Eagle Medium (DMEM) media without fetal bovine serum and antibiotics. After treatment, cells were fixed using 4% PFA and blocked using 10% NGS. The primary antibodies for acetylated tubulin (Sigma Catalog no. T7451) and tyrosinated tubulin (Sigma Catalog no. MAB1864) were used to stain stable and dynamic microtubules, respectively. For secondary antibodies, Alexa flour 594 goat anti-mouse (Invitrogen Catalog no. A11032) and Alexa flour 647 goat anti-rat (Invitrogen Catalog no. A21247) were used for acetylated tubulin and tyrosinated tubulin labeling, respectively. For COS7, ~20 cells and for HEK293T, ~10 cells were imaged per condition in each experiment for quantitative analysis. Fiji was further used to process the images, and for quantification, the stable to dynamic microtubule ratio was calculated by manually tracking microtubules in each NAV3 $^{WT}$ and variants over-expressing cells. Paired t-test was applied to assess the microtubule stability difference among the nocodazole-treated variants and WT-transfected cells.

## Morpholino-based *nav3* knockdown in zebrafish

To study the in vivo impact of *nav3* knockdown, we used Tg(*nrd:egfp*) transgenic zebrafish line. Previously reported[12], ATG blocker (5′-TTGAAGCAACACCAACTACCGGCAT-3′) morpholino (ATG_MO 8 ng & 10 ng) and splice site blocking (5′-TGTATGAGCTCTCACCTAT-TATCTG-3′) morpholino (SS_MO 10 & 12 ng) were used for injecting 1-2

cell stage embryos. For controls, scrambled nucleotide morpholino was used (5′-CCTCTTACCTCAGTTACAATTTATA-3′). Embryos were phenotypically evaluated at the 4th day post fertilization (dpf). The head to body length ratio was measured for each larva in the mild and moderate class of controls and morphants to measure their head size. Both dorsal and lateral view images for each larva were taken using the same acquisition settings for overall brain and length measurements. Brain size for each individual larvae was measured and divided by its body length to avoid discrepancy based on the overall size of the larvae among each group. To check the splicing impact of SS_MO, injected morphants were collected to extract total RNA to form cDNA. Primers were designed from exons 1 and 4 (Forward: 5′-GTTGCTTCAAAACTCCGACAG-3′; Reverse: 5′-GAGAGCCCTTGAA-CACTGACA-3′) to determine its effect on exon skipping or intron retention.

Further, to assess neuronal pattern, embryos at 4th dpf were treated with 1-Phenyl-2-thiourea (PTU) 0.2 mM (TCI Catalog no. P0237), fixed in 4% PFA overnight at 4 °C and mounted in 1% low melting agarose (NuSieve Catalog no. 50080). Confocal imaging was performed using Nikon spinning disk W1 with a 20X lens and the same acquisition settings for pinhole, exposure, and gain. The images are further processed through Fiji software to develop 3D structure using maximum intensity. The represented images in Fig. 5g are dorsal snapshots of 3D developed images. Student *t-test* was used to evaluate differences in various groups.

## Behavioral assessment of zebrafish morphants using light as a stimulus

At 5th dpf, each embryo activity was measured by distance traveled after light stimulus using ZebraLabs System. For this assay, each larva was singly housed in 48 well plates in 400 μl E3 medium an hour before the experiment for acclimation. The embryos were then subjected to 5 min in the dark, followed by 5 min in the light as an abrupt stimulus. The sensitivity threshold was set to 20 with the maximum burst threshold of 50. The average distance traveled by each larva in 10 min was plotted on a violin graph. The difference in activity of all groups was assessed using the Paired t-test.

## Phenotypic rescue studies in zebrafish morphants using human full length *NAV3* mRNAs

For rescue and functional analysis of ID-associated variants in zebrafish morphants, we cloned *NAV3*$^{WT}$ cDNA in the pT7TS vector after linearizing

the vector with the *BGI II* restriction enzyme. Site directed mutagenesis using WT construct as a template was performed to introduce *NAV3* variants found in our cohort. After confirming through Sanger sequencing, *NAV3* constructs were linearized using *NDEI* restriction enzyme, and mRNAs were synthesized using the HiScribe T7 Quick RNA synthesis kit (New England Biolabs Catalog no. E2050S). A dosage response was performed to optimize the mRNA dose using the $NAV3^{WT}$ mRNA at 250 pg, 500 pg, 750 pg, and 1 ng co-injected with 8 ng ATG_MO, and subsequent studies used 500 pg of $NAV3^{WT}$ and variants harboring mRNAs co-injected with ATG_MO for rescue studies in zebrafish larvae.

## Data availability

All data is mentioned in the main text and Supplementary Figures. All the source data is given in the Supplementary Data 1 file, available to readers.

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

## Acknowledgements
We would like to thank families and enrolled subjects for their participation in the study. We would also like to thank Dr. Asma. A. Khan, Asaad Usmani, Sehar Riaz, Sakina Rehman, and Amna A. Zaib for helpful discussion and technical support, and Dr. Joseph Mauban and Confocal Core Facility, University of Maryland, Baltimore for their guidance and technical assistance. For the PE30 family, the exome was done by Norwegian High-Throughput Sequencing Centre, supported by the "Functional Genomics" and "Infrastructure" programs of the Research Council of Norway and the Southeastern Regional Health Authorities. We would also like to thank UNINETT Sigma2 for data analysis and storage. This work was supported in part by R01NS107428 (S.R.) from NINDS/NIH and The Norwegian National Advisory Unit on Rare Disorders. M.Z. receives research support from the German Research Foundation (DFG 458949627; ZE 1213/2-1). M.Z. acknowledges grant support by the EJP RD (EJP RD Joint Transnational Call 2022), the German Federal Ministry of Education and Research (BMBF, Bonn, Germany), awarded to the project PreDYT (PREdictive biomarkers in DYsTonia, 01GM2302), by the Federal Ministry of Education and Research (BMBF) and the Free State of Bavaria under the Excellence Strategy of the Federal Government and the Länder, as well as by the Technical University of Munich—Institute for Advanced Study. The work was also supported by the National Institute for Neurological Research, Czech Republic, Programme EXCELES, ID Project No. LX22NPO5107, funded by the European Union—Next Generation EU and also by the Charles University: Cooperation Program in Neuroscience.

## Author contributions
Sh.R. and S.R. designed and organized the study. A.G. and T.A. collated and composed sections describing human clinical data, generated constructs, performed in vitro and zebrafish-related studies. T.A. generated the mutant constructs. and analyzed zebrafish-related data and edited figures; A.G., Z.M.A., and S.R. composed the core manuscript. Sh.R., Z.M.A., and S.R. supervised and validated data and reviewed and edited the manuscript. P.S., D.M., A.K., E.F., M.U., B.I., B.C., A.L.B., A.S., A.A.K., P.K., C.C., A.M.I., M.Z., M.B., P.H. R.J., all contributed clinical patient information. All authors read and provide comments to the final manuscript and approved the submission.

## Competing interests
The authors declare no competing interests.

## Ethics
Written informed consent, including specific consent to use photographs, was obtained for all individuals involved. This study adheres to the World Health Association Declaration of Helsinki and was approved by the Institutional Review/Ethic Boards of all the participating institutes. All zebrafish-related experiments were conducted in accordance with recommendations of the Guide for the Care and Use of Laboratory Animals of the University of Maryland Baltimore, protocol 0420002.

## Additional information

[1]Department of Otorhinolaryngology-Head & Neck Surgery, School of Medicine University of Maryland, Baltimore, MD, USA. [2]Centre of Excellence in Molecular Biology, University of the Punjab, Lahore, Pakistan. [3]Division of Pediatric and Adolescent Medicine, Oslo University Hospital and University of Oslo, Oslo, Norway. [4]Department of Medical Genetics, Oslo University Hospital and University of Oslo, Oslo, Norway. [5]Faculty of Biological Sciences, Department of Zoology, University of Lakki Marwat, 28420 Khyber, Pakhtunkhwa, Pakistan. [6]Institute for Medical Genetics and Applied Genomics, University of Tübingen, Tübinge 72076, Germany. [7]Alexander von Humboldt Fellowship Foundation, Berlin 10117, Germany. [8]Department of Life Sciences, School of Science, University of Management and Technology, Lahore, Pakistan. [9]Nantes Université, CHU Nantes, Service de Génétique Médicale, 44000 Nantes, France. [10]INSERM UMR1231 GAD "Génétique des Anomalies du Développement", FHU-TRANSLAD, Université de Bourgogne Franche-Comté, Dijon, France. [11]National Center of Genetics (NCG), Laboratoire national de santé (LNS), 1, rue Louis Rech, L-3555 Dudelange, Luxembourg. [12]Department of Pediatrics, CHU de Nice, Fondation Lenval, Nice, France. [13]Department of Medical Genetics, University of Calgary, Calgary, Alberta, Canada. [14]Institute of Neurogenomics, Helmholtz Munich, Neuherberg, Germany. [15]Institute of Human Genetics, Technical University 2 of Munich, School of Medicine, Munich, Germany. [16]Institute for Advanced Study, Technical University of Munich, Lichtenbergstrasse 2 a, D-85748 Garching, Germany. [17]First Department of Neurology, Faculty of Medicine, St. Anne's University Hospital, and CEITEC, Masaryk University, Brno, Czech Republic. [18]Department of Neurology, Charles University, First Faculty of Medicine and General University Hospital in Prague, Prague, Czech Republic. [19]Jinnah Burn and Reconstructive Surgery Centre, Allama Iqbal Medical Research, University of Health Sciences, Lahore, Pakistan. [20]These authors contributed equally: Amama Ghaffar, Tehmeena Akhter.
✉e-mail: sriazuddin@som.umaryland.edu

