## [Peer Review File · Communications Biology]

Reviewers' comments:

Reviewer #1 (Remarks to the Author):

Ghaffar et al. examined several Human families exhibiting intellectual disability, brain morphology, and other behaviors and identified their mutations in NAV3. They characterized these mutations in silico, in vitro, and in zebrafish vivo model and showed that they cause impaired microtubule stability, microcephaly, and behavioral defects. This work is valuable because it uses multiple patient information and suggests the causative gene and functions by experiments using model organisms. However, there are significant shortcomings in their data, which need to be improved. Also, the author did not indicate the novelty of this work, which needs to be updated. Listed below are my specific comments.

- In this manuscript, the authors mostly enumerated and compared the mutations they identified. The authors should provide careful comparative explanations of what is different and significant compared to previous findings of the Nav3 mutation sites. Currently, the fundamental question and the significance of this work are not clear.

- Line 151-154: The authors wrote, "...NAV3 also showed overlapping expression with stable microtubule structures along with condensed granule expression towards +Tips of microtubules. Moreover, the protein truncating variants, except p. Asp1208Argfs*58, did not show +Tip granule formations". However, it is difficult to conclude these results from the images shown in Figure 3. The enlarged images should be shown to explain these tendencies (overlapping expression and condensed granule expression). Also, the authors should use arrows and arrowheads to indicate these points in the Figures.

- Figure1d : MRI images of control individuals should also be shown to compare and indicate the defects.

- Figure 3: The order of the images should be improved for clarity. p.Ser1325Asn and pArg2261Cys should be moved to the right side of NAV3wt, pEGFP, and Mock.

- Figure4a: The authors need to show the diagram of the expected protein and defects in its function with the splice MO. Also, this needs to be discussed in explaining Figure 5C data.

- Line 200, 205 etc: "while highly significant ($p < 0.0001$) differences between controls vs nav3 ATG-MO were found"

The authors often wrote "significant differences" without explaining the details. For clarity, more explicit expressions such as "significantly increased" or "significantly decreased" should be used.

- Discussion line 240-258: The general explanation is too long. It should be shortened or moved to the introduction part and changed to a discussion based on the results of this study.

- Some of the figure numbers in the text are incorrect. Please correct them.

Reviewer #2 (Remarks to the Author):

In the current manuscript, Ghaffar et al. report a spectrum of neurodevelopmental disorders in individuals with biallelic, de novo or heterozygous pathogenic variants of NAV3 gene. The spectrum includes microcephaly, developmental delay, facial malformations, intellectual delay etc. Further, the authors checked microtubule stability, a known function of NAV3, in COS cells and modelled

microcephaly, anomalies in body growth and behavior using zebrafish and morpholino assays. Overall, it is an interesting study, especially with the addition of the new clinical cohort. However, followed issues need to be addressed:

1. Figure legends do not match the figures, in multiple instances. This needs to be rectified.
2. Instead of using human cell lines like HEK293 or Sh-SY5Y, use of COS7 cells is not clear. Also, currently interpreting clinical or cellular processes using a single cell line is not recommended. So, it would be useful to reproduce the microtubule data in other cell lines.
3. Conceptually, NAV3 is known to be involved in microtubule dynamics, and mutations are supposed to disrupt that. So, is the cell line experiment just a proof of principle in their study, or does it have greater implications?
4. The +tips of the microtubules are not observable in Figures 3 and 4. Higher magnification and insets would be necessary for better demonstration.
5. The motor problems in morpholino-treated zebrafish mutants cannot be extrapolated to modeling cognition, unless involved neural circuits are dissected out. Possibly the language used in the manuscript needs to be altered, highlighting this as only an empirical data with available systems. If a murine model is available, behavioral tests of different kinds are better be performed there, but it is understandable that authors may not have access to it.

6. Line 275-277: "In conclusion, we present genetic and functional evidence to support the role of NAV3 in brain development and function in humans and zebrafish through the regulation of microtubule dynamics."

This statement is too extrapolated based on the data produced in this paper. No functional evidence has been provided for humans (the clinical report of phenotype can only be considered if clinical assays to assess intellectual delay etc are provided as part of this paper). The cell line used is also not originated from human tissue. In the zebrafish model, only one sensory behavior was tested. In my opinion, multiple assays need to be performed on both models used before arriving at this inference.

Reviewer #3 (Remarks to the Author):

The manuscript entitled 'Variants of NAV3, a neuronal morphogenesis protein, cause intellectual disability, developmental delay and microcephaly' reports mutations in NAV3 in individuals across continents leading to intellectual disability and microcephaly. In a bid to understand the mechanisms, the authors use cell lines and zebrafish larvae expressing the pathogenic variants and show impaired microtubule stabilization, brain defects and motor deficits in these models.

The authors show using data from 7 families that NAV3 mutations lead to ID. Their work complements a previous screen which led to the discovery of NAV3 mutations in autism spectrum disorder (Zhou et al, 2022). Though the numbers are still few likely due to the prevalence of these mutations, the efforts of the authors to collect data from across ethnicities, in populations in different continents and profile the mutations is commendable, I congratulate them for their discovery.

My criticism is primarily to strengthen the mechanistic second part of the study, and are as follows.

1. Figure 2C – the zebrafish sequence is likely shifted and overlaps with the frog sequence
2. The current description of Figure 2D does not give new insights into the functioning of the variants. This data can be better explained to speculate on the altered functioning of these mutant proteins.
3. NAV3 is highly expressed during neurodevelopment and plays a crucial role in this process.

However, it is surprising to note that neuro progenitor cells in the scRNA-seq data analysis do not express NAV3. Can the authors comment on this?

4. I couldn't find details of the control morpholino.

5. The authors describe defective 'neuronal patterns' in the optic tectum of NAV3 morphant brains and have quantified optic tectum area. Both these measures could be different by staining sections from different levels of the brain in the Z-axis. Indeed, the eye size in the images presented does appear different and to be from different levels, and I believe it is not permissible to compare such sections.

6. For the experiments on locomotor activity, were the defects caused by the severe defects the authors observed in the morphants? How did the authors control for this?

7. Similar to point 6, it might be misleading to measure brain to body length ratio due to the defects observed in the morphants. In this case, one suggestion would be to image larval heads using the same acquisition settings and compare brain volume.

8. Though the authors do demonstrate that the NAV3 mutations affect cells in culture and zebrafish, some mechanistic detail, for instance, details of the neuronal subtypes affected in the morphants could help form a hypothesis on the mode of action, in addition to validating the pathogenicity of these variants.

Reviewer #1

Q1: In this manuscript, the authors mostly enumerated and compared the mutations they identified. The authors should provide careful comparative explanations of what is different and significant compared to previous findings of the Nav3 mutation sites. Currently, the fundamental question and the significance of this work are not clear.

Answer: Thanks. We have added the discussion section as follow to highlight the similarities and differences of two studies:

Pages 10-11: Lines: 226-249:

“In this study, we describe a cohort of patients harboring deleterious variants in the *NAV3* gene. The affected individuals show ID, microcephaly, skeletal deformities, ocular anomalies, and behavioral issues besides other clinical symptoms. Recently, another study documented nine nonsense, 12 frameshifts, and three splicing site predicted pathogenic variants of *NAV3* in subjects with ASD and NDDs (Fig. 7) based on transmission disequilibrium test¹³. ASD in about half of their cases coexisted with ID (18/35) and attention-deficit/hyperactivity disorder (ADHD; 15/35)¹³. Among our cases, ADHD was observed in subjects with truncating variants (Table 1). In contrast to almost all truncating variants found in ASD cases, half of our subjects with ID had missense variants (Fig. 1). Apparently, missense variants (presumably hypomorphic alleles) of *NAV3* cause ID, while truncating alleles are responsible for ASD with additional neurological findings. In general, phenotype spectrum observed in our cohort overlaps with the prior study, and thus further confirm the role of *NAV3* in the brain development and cognitive function.

In contrast to prior study, which mostly report *de novo* variants, among our cases, we observed three different allele patterns, autosomal recessive, dominant, and *de novo*. Although our sample is not large enough for a meaningful genotype-phenotype correlation, intriguingly, individuals with biallelic *NAV3* variants had severe ID with the prevalence of other features such as microcephaly, skeletal deformities, and behavioral problems. Affected individuals with heterozygous alleles (dominant and *de novo* cases) had mild ID with less prevalence of subsided issues (Table 1), which might reflect haploinsufficiency. However, larger variant sets and functional studies would be required to classify disease mechanisms of variants and to devise clinically applicable genotype-phenotype correlations.”

Q2: Line 151-154: The authors wrote, "...NAV3 also showed overlapping expression with stable microtubule structures along with condensed granule expression towards +Tips of microtubules. Moreover, the protein truncating variants, except p.Asp1208Argfs*58, did not show +Tip granule formations". However, it is difficult to conclude these results from the images shown in Figure 3. The enlarged images should be shown to explain these tendencies (overlapping expression and condensed granule expression). Also, the authors should use arrows and arrowheads to indicate these points in the Figures.

Response: Thanks for the helpful suggestion, we have updated the figures 3 and added a new figure 4 with zoomed in images and arrowheads to pin point granule formation at +ive tip ends.

Q3: Figure1d: MRI images of control individuals should also be shown to compare and indicate the defects.

Response: Done. We have added the MRI images of a control subject in the figure 1d for comparison.

Q4: Figure 3: The order of the images should be improved for clarity. p.Ser1325Asn and p.Arg2261Cys should be moved to the right side of NAV3wt, pEGFP, and Mock.

Response: Thanks. We have updated the figure3, accordingly.

Q5: Figure4a: The authors need to show the diagram of the expected protein and defects in its function with the splice MO. Also, this needs to be discussed in explaining Figure 5C data.

Response: Previous figure 4, is now figure 5 in the revised manuscript. We have added a schematic in figure 5c to illustrate the impact of splice MO. Also, the information has been added in the results section as follow:

Page 9; lines 194-197:

“SS_MO injected morphants amplified cDNA as compared to control resulted into exon 2 skipping (c.del178_195), and deletion of 118bp from *nav3* mRNA, leading to reading frameshift and premature truncation (p.Ile60Leufs*4) of the encoded protein (Fig. 5c).”

Q6: Line 200, 205, etc.: "while highly significant ($p < 0.0001$) differences between controls vs. *nav3* ATG-MO were found." The authors often wrote "significant differences" without explaining the details. For clarity, more explicit expressions such as "significantly increased" or "significantly decreased" should be used.

Response: Thanks. The “significance” description overall in the manuscript has been rephrased clearly to state if the variants have “significantly increased” or “decreased” expression.

Q7: Discussion line 240-258: The general explanation is too long. It should be shortened or moved to the introduction part and changed to a discussion based on the results of this study.

Response: Based on your suggestions, we have revised the discussion section and removed general explanation.

Q8: Some of the figure numbers in the text are incorrect. Please correct them.

Response: Sorry for the errors. We have updated and confirmed the figure numbers in the revised manuscript.

Reviewer #2

Q 1: Figure legends do not match the figures in multiple instances. This needs to be rectified.

Response: Sorry for the errors. We have updated the figures and figure legends in the revised manuscript to address the concern.

Q 2. Instead of using human cell lines like HEK293 or Sh-SY5Y, use of COS7 cells is not clear. Also, currently, interpreting clinical or cellular processes using a single cell line is not recommended. So, it would be useful to reproduce the microtubule data in other cell lines.

Answer: Thanks for the suggestion. We have reproduced the microtubule data in HEK293T cells, and have added a new Figure 4 in the revised manuscript. We used COS7 cells due to the broad structure of cell type to easily visualize subcellular localization and changes in microtubule structure in the presence of microtubule inhibiting drug, as it's a primate species like humans with a similarity index of 98%. Essentially, similar to COS7 cells, we overexpressed NAV3 WT or variants harboring constructs, in the presence of nocodazole. Like COS7 cells, HEK293T heterologous cells overexpressing WT-NAV3 showed microtubule stability. However, this impact was greatly reduced for variants expressing NAV3 proteins.

Q 3. Conceptually, NAV3 is known to be involved in microtubule dynamics, and mutations are supposed to disrupt that. So, is the cell line experiment just a proof of principle in their study, or does it have greater implications?

Answer: Yes, the cell line experiments were performed to determine if the ID-associated variants identified in our study are “benign” or have “pathogenic/deleterious” impact on the NAV3 function. We choose microtubule dynamics assay, as it is known that NAV3 has a role in this process. Our cell line studies both in COS7 and HEK293 cells confirmed the deleterious impact of all the identified NAV3 variants.

Q 4. The +tips of the microtubules are not observable in Figures 3 and 4. Higher magnification and insets would be necessary for better demonstration.

Answer: Thanks. We have updated figures 3 and 4 with higher magnification insets, as suggested.

Q 5. The motor problems in morpholino-treated zebrafish mutants cannot be extrapolated to modeling cognition unless involved neural circuits are dissected out. Possibly, the language used in the manuscript needs to be altered, highlighting this as only empirical data with available systems. If a murine model is available, behavioral tests of different kinds are better be performed there, but it is understandable that authors may not have access to it.

Answer: We agree, and have updated the text as follow:

Page 13, Lines 287-296:

“The light-based stimulation analysis shows low movements of *nav3* morphants (Fig. 5i), which could be attributed to one or more of the following: a) generalized developmental and structural abnormalities; b) brain developmental deficits, or c) heart deficits and yolk edema. The optic tectum is center of neurons receiving signals from the retina and torus longitudinalis for vision-based responses,³¹ and low movement response in *nav3* morphant might be due to structural or connectivity deficits in these regions of the brain. However, our light-based simulation analysis is only empirical data, and cannot be used to define cognition function. Future studies in *Nav3* murine model including behavioral assays, neuronal connectivity, and physiological measurement would help in defining the precise role of NAV3 in cognitive function.”

Q 6. Line 275-277: “In conclusion, we present genetic and functional evidence to support the role of NAV3 in brain development and function in humans and zebrafish through the regulation of microtubule dynamics.”: This statement is too extrapolated based on the data produced in this paper. No functional evidence has been provided for humans (the clinical report of phenotype can only be considered if clinical assays to assess intellectual delay, etc, are provided as part of this paper). The cell line used is also not originated from human tissue. In the zebrafish model, only one sensory behavior was tested. In my opinion, multiple assays need to be performed on both models used before arriving at this inference.

Answer: Thanks. We have rephrased the sentence as follow in the revised manuscript:

Pages 13-14, Lines 297-300:

“In conclusion, taken together with prior studies, biallelic and mono-allelic variants of *NAV3* are responsible for a spectrum of neurodevelopmental disorders with clinical features including ID, microcephaly, global developmental delay, or autism. Our findings further substantiate association of *NAV3* variants with NDDs in humans.”

Reviewer #3:

Q 1: Figure 2C – the zebrafish sequence is likely shifted and overlaps with the frog sequence.

Response: The panel C for Figure 2 has been updated with proper alignment.

Q 2: The current description of Figure 2D does not give new insights into the functioning of the variants. This data can be better explained to speculate on the altered functioning of these mutant proteins.

Response: Thanks. We have updated the description of protein modeling data as follow:

Page 7, lines 145-155:

“For hydrogen bond analysis PyMOL was used to find interacting bonds of WT and mutated amino acids to neighboring amino acids. The p.Ser1326Asn variant is predicted to cause a loss of hydrogen bond with p.Ser1128 residue due to small size and less hydrophobic nature of asparagine residue as compared to WT serine residue, and thus might impact protein secondary structure. The p.Ser1681Arg variant is also predicted to remove hydrogen bonding with p.Glu1734 residue due to less hydrophobic, smaller size and positive charge of arginine residue (Fig. 2d). Moreover, p.Ser1681Arg substitution causes energy destabilization (-0.0 kcal/mol). In contrast, the p.Arg2261Cys variant resulted in loss of four hydrogen bonds, two with p.Arg2264 and two with p.Glu2320 residues (Fig. 2d), thus likely impact protein folding and secondary structure.”

Q 3: NAV3 is highly expressed during neurodevelopment and plays a crucial role in this process. However, it is surprising to note that neuro progenitor cells in the scRNA-seq data analysis do not express NAV3. Can the authors comment on this?

Response: We speculate that nav3 is involved in the early morphogenesis of the neuronal cells as it's a +ive tip protein that is involved in the proper migration of differentiated cells, and that's the probable reason for nav3 high expression in post-mitotic neurons as compared to neuronal progenitor cells where cells are involved in neurogenesis, but migration is limited.

Q 4: I couldn't find details of the control morpholino.

Answer: Control morpholino's are scrambled nucleotides readily available from Gene Tools, LLC with Standard sequence 5` CCTCTTACCTCAGTTACAATTTATA 3`, used for in vivo control group experiments. We have added this information in the revised manuscript method section.

Q 5: The authors describe defective 'neuronal patterns' in the optic tectum of NAV3 morphant brains and have quantified optic tectum area. Both these measures could be

different by staining sections from different levels of the brain in the Z-axis. Indeed, the eye size in the images presented does appear different and to be from different levels, and I believe it is not permissible to compare such sections.

Response: We agree that comparing different levels of the brain in the Z-axis would provide different measures. However, we imaged both control and *nav3* **mild** class morphants using confocal Nikon spinning disk W1 with the same acquisitions for pinhole, gain, and exposure. The *nav3* ATG_MO injected morphants on *neuroD* background, had generalized microcephaly (Fig. 5f), and essentially all regions of brain including optic tectum, the habenula, cerebellum, midbrain and hindbrain were reduced in size and thus could not be used as reference. However, we did use overall body length as reference in our quantitative analysis, and all the data plotted was normalized against body length between control and *nav3* morphants. Moreover, as shown in the bright light images in Fig. 5b, the overall eye size is smaller in *nav3* morphants as compared to controls, which corroborate with confocal images in Fig. 5g. Optic tectum measurements were included as an example and for quantitatively data analysis of smaller brain size in fig. 5h.

Q 6: For the experiments on locomotor activity, were the defects caused by the severe defects the authors observed in the morphants? How did the authors control for this?

Response: We agree with the reviewer that severe and even moderate class *nav3* morphants have overall body deficits, which could impact locomotor activity. Therefore, we only used “mild and normal class” morphants in our behavior assay (as stated in the revised manuscript lines 211-212).

Q 7: Similar to point 6, it might be misleading to measure brain to body length ratio due to the defects observed in the morphants. In this case, one suggestion would be to image larval heads using the same acquisition settings and compare brain volume.

Response: We agree, and we only used “mild class” morphants, that have essentially similar body length when compared to normal larvae, for our brain to body length ratio study. We imaged dorsal and lateral side of larvae under same acquisition settings.

Q 8: Though the authors do demonstrate that the NAV3 mutations affect cells in culture and zebrafish, some mechanistic details, for instance, details of the neuronal subtypes affected in the morphants, could help form a hypothesis on the mode of action, in addition to validating the pathogenicity of these variants.

Response: We agree with the reviewer that detailed analysis of neuronal subtypes would help in developing hypothesis and mechanistic insights. However, due to generalized microcephaly and gross developmental deficits observed in *nav3*

morphants, it would be difficult to distinguish primary vs secondary impact on the neuronal pattern due to overall growth issues. We essentially observed smaller size of all the major areas of the brain. We believe that further studies in murine model with conditional deletion of *Nav3* in the brain only, would be more mechanistically insightful and helpful in determine the precise role of NAV3 in the neuronal growth, patterning, and connectivity, and thus these studies are part of our further direction of the research project in the lab. However, these studies would require at least 1.5-2 years to complete, and are beyond the scope of this study. We have added the following sentences in the revised discussion section:

Pages 12-13, lines 273-286:

“In our zebrafish *nav3* knockdown model, we observed major developmental deficits, including heart malformation and microcephaly (Fig. 5b), similar to zebrafish *nav3* knockouts¹¹. We observed generalized microcephaly in *nav3* morphants as well. The major brain structures morphologically impacted in *nav3* morphants were a) habenula (Hb), which contains habenular neuronal cell types (Hb01-Hb15); b) torus longitudinal (TL); and c) Optic tectum (OT) carry mostly granule cells with multiple neuronal subtypes, including mostly GABAergic neurons; and d) cerebellum, which harbor Purkinje cells, eurydendroid cells, and granule cells with long axon terminals and dendrites. All these regions of brain are functionally interconnected for coordinated motor responses. Thus, reduced sizes of these major structures among *nav3* morphants might impact the neuronal communication among different parts of the brain involved in motor responses. Further research in murine models with conditional deletion of *Nav3* in specific brain regions would aid in determining the precise role of NAV3 in neuronal subtype proliferation, patterning, connection, and communication.”

REVIEWERS' COMMENTS:

Reviewer #1 (Remarks to the Author):

The author responded to my comments, and the manuscript was nicely improved.

Reviewer #2 (Remarks to the Author):

Minor comments:

The insets of microtubule dynamics look like just zoomed-in version of the same magnification shown in the full images. These are quite unclear and do not help visualizing anything at a deeper level. It is recommended to take inset images at a much higher magnification than used for imaging the whole cell.

Reviewer #3 (Remarks to the Author):

The authors have addressed the comments and concerns of all the reviewers fairly well.

Reviewer #1

The author responded to my comments, and the manuscript was nicely improved.

Answer: Thank you so much.

Reviewer #2

Minor comments:

The insets of microtubule dynamics look like just zoomed-in version of the same magnification shown in the full images. These are quite unclear and do not help visualizing anything at a deeper level. It is recommended to take inset images at a much higher magnification than used for imaging the whole cell.

Response: Thanks. Based on your suggestion, we have revised the figure 3 with higher resolution images.

Reviewer #3:

The authors have addressed the comments and concerns of all the reviewers fairly well.

Response: Thank you so much.